# Redesign of Leg Assembly and Implementation of Reinforcement Learning for a Multi-Purpose Rehabilitation Robotic Device (RoboREHAB)

**Jacob Anthony [1], Chung-Hyun Goh [1,*] , Alireza Yazdanshenas [1] and Yong Tai Wang [2]**

[1] Department of Mechanical Engineering, University of Texas at Tyler, Tyler, TX 75799, USA; janthony6@patriots.uttyler.edu (J.A.); ayazdanshenas@patriots.uttyler.edu (A.Y.)

[2] Rochester Institute of Technology, College of Health Sciences and Technology, Rochester, NY 14623, USA; ytwchst@rit.edu

\* Correspondence: cgoh@uttyler.edu; Tel.: +1-903-566-6125

**Abstract:** Patients who are suffering from neuromuscular disorders or injuries that impair motor control need to undergo rehabilitation to regain mobility. Gait training is commonly prescribed to patients to regain muscle memory. Automated-walking training devices were created to aid this process; while these devices establish accurate ankle-path trajectories, the knee and hip movements are inaccurate. In this work, a redesign of the leg assembly in a multi-purpose rehabilitation robotic device (RoboREHAB) was explored to improve hip- and knee-movement accuracy by adding an extra link and rollers to the assembly. Motion analysis was employed to test feasibility, reinforcement learning was utilized to train the new leg assembly to walk, and the joint motions achieved with the redesign were compared to those achieved by motion-capture (mocap) data. As a key result, the motion analysis showed an improvement in the knee- and hip-path trajectories due to the added roller/joint segment. The redesigned leg assembly, under the reinforcement-learning policy, showed a 5% deviation from the motion-capture joint trajectories with a maximum deviation of 51.177 mm but maintained a similar profile to the mocap trajectory data. This is an improvement over the original two-segment design, which achieved a maximum deviation of 72.084 mm. These results in the knee- and hip-joint movements more closely reflect the mocap and motion-analysis results, validating the redesign and opening it up to further experimentation and technical improvement.

**Keywords:** gait motion; leg-assembly design; motion capture (mocap); reinforcement learning; multi-purpose rehabilitation robotic device (RoboREHAB); feasibility analysis

## 1. Introduction

As the age of patients increases, so do the chances of them suffering disabilities. Patients who have suffered from stroke, spinal cord injury (SCI), or brain trauma usually develop a disability that affects motor function. SCIs can be physically and emotionally damaging for the patient and treatment can cost up to USD 4.6 million [1]. Neuromotor deficits are also present in patients with cerebral palsy (CP), a medical condition that can impair the patient's abilities to walk or keep their balance [2]. To help CP patients regain motor function, muscle training and exercises such as constraint-induced movement therapy have been used to improve the function of the affected limb [3–5]. However, there is speculation about whether the patient is regaining control of their limb, or the patient is using an unnatural movement to compensate for the loss of motor control [5]. This has led to an increase in research dedicated to developing rehabilitation methods to help patients regain motor control. Gait training is a rehabilitation strategy to help patients regain motor control in their lower extremities by facilitating exercises mimicking a natural walking pattern. Recent advancements in technology have inspired biomedical research to improve gait-training effectiveness to help the patient make a full recovery. One such

solution is the use of powered robotic exoskeletons to aid the patients in the gait-training exercises. For example, one study used a lower-limb exoskeleton to assist in the knee extension of CP patients with crouch gait [6]. Some studies have been made to test the safety and feasibility of exoskeletal devices to treat patients suffering from complete and incomplete SCI and how this treatment affects their gait [7]. For instance, the Ekso mobile exoskeleton demonstrated an improvement in gait function for SCI patients; however, the donning and doffing of the exoskeleton is a challenge for the patients [8]. Research conducted at the University of Texas at Tyler involves designing and developing a device called RoboREHAB, a multi-purpose rehabilitation robotic device to correct gait patterns for walking and facilitate aid in sit-to-stand motions [9]. Within this work, motion-capture (mocap) technology was used to record the motion path of the ankle when walking on a treadmill. The path trajectory for the ankle joint is presented in Figure 1a. These mocap data were then used to derive the path trajectories for the knee and hip joints as plotted in Figure 1b,c [9]. The original design of the RoboREHAB leg assembly shown in Figure 2 does not accurately replicate the path trajectories of the knee and hip joints shown in Figure 1b,c due to a lack of degrees of freedom (DOF) in the assembly. In this paper, the RoboREHAB leg assembly was redesigned to improve the path trajectories on the ankle, knee, and hip joints by adding an additional link on the knee joint. This results in increased DOFs such as pitch moment on the knee joint during the gait-motion simulations so that it produces better path trajectories on the joints than the original design, in which only swing motion was allowed on the knee joint.

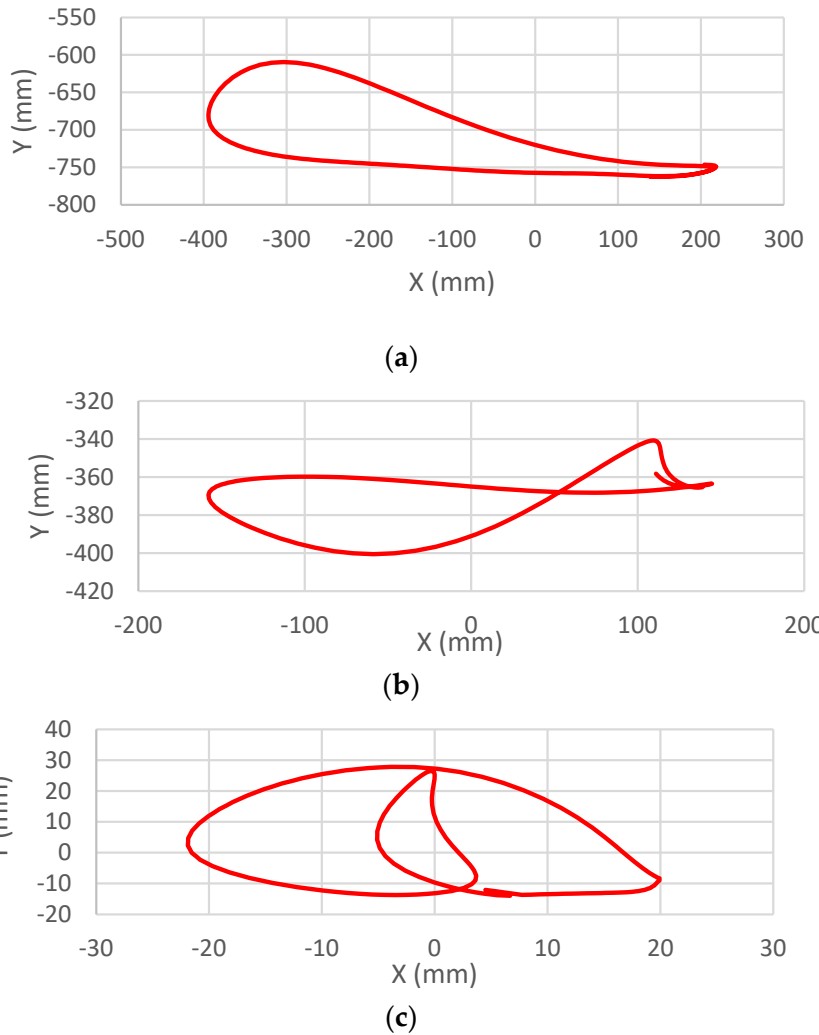

**Figure 1.** Ideal joint-motion paths for the (**a**) ankle, (**b**) knee, and (**c**) hip joint.

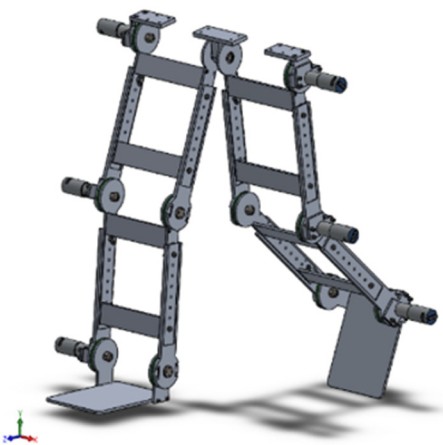

**Figure 2.** Original RoboREHAB leg assembly design.

Machine-learning solutions have also been researched and developed to aid patients in regaining motor function [10–12]. One solution in the realm of machine learning is reinforcement learning, a technique that trains an agent to analyze a given pose/state of a subject and suggests actions for the subject to execute to maximize the reward that it receives [13–15]. The agent is usually trained using an 'actor–critic' network. This agent has an 'actor' that performs the action suggested by the agent and a 'critic' associates a reward to the action performed. The reward comes in two forms: the individual episode reward, R, and the long-term discounted reward that is initially expected from the action, Q. Two common types of Q-learning algorithms are the deep-deterministic policy gradient (DDPG) and the twin-delayed DDPG (TD3) algorithms. DDPG algorithms develop a single Q-function when training the agent [16]. However, DDPG algorithms tend to overestimate Q-values, which can cause the algorithm to break the policy that it has learned [17]. TD3 algorithms rectify this issue by developing two Q-functions and preventing the algorithm from exploiting the Q-function by updating the policy less frequently and adding noise to the target action [17,18]. The above-mentioned reinforcement-learning technique has been used to make a lower-limb exoskeleton follow a natural walking gait. This allows for a more natural gait to be developed by the patient when using the exoskeleton, but most designs do not have the degrees of freedom needed to replicate the joint-path trajectories accurately [19]. In this work, a redesign of the leg assembly of the RoboREHAB is made to improve the kinematics at the knee joint to best replicate the mocap data collected. A reinforcement-learning agent is trained in Simulink with the aim to have a model of the leg assembly follow the mocap data. The original RoboREHAB leg assembly needs to be modified by adding additional DOFs to the leg assembly and trained to follow the mocap data using reinforcement learning. This modified leg assembly will then be compared to the original RoboREHAB leg assembly based on the path trajectories achieved. The path trajectories achieved by the modified assembly and the original assembly will be compared. This work will help patients with diminished motor control recover with a more accurate gait cycle.

## 2. Design and Methods

The workflow for this work is demonstrated in Figure 3. The 2022 version of Solid-Works software was used to redesign the leg assembly and to establish feasibility through motion analysis. MATLAB 2022a was used to create a MATLAB/Simscape (Simscape) model that reflects the proposed redesign of a leg composed of three links (upper leg, lower leg, and foot). Reinforcement learning was implemented to train the new leg assembly to follow the mocap data. These results were compared to SolidWorks and mocap data for validation purposes.

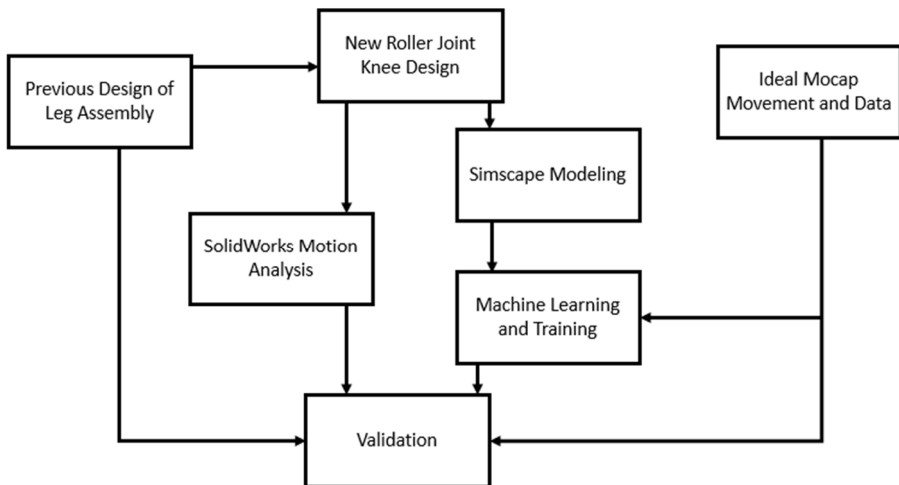

**Figure 3.** Workflow for this study.

The original design of the RoboREHAB leg assembly has the upper-leg section connected to the lower-leg section as a fixed knee joint. The human knee joint is not so much a fixed joint, but rather, a saddle joint that the femur and the tibia form, as seen in Figure 4. The saddle joint allows the proximal end of the tibia to roll and travel along the distal end of the femur. By this logic, the interaction between the ends of the femur and tibia can be considered a small segment with two rollers in contact at the top and bottom of the segment. To emulate this movement, the redesign of the leg assembly involved the addition of a link between the upper- and lower-leg sections, along with two rollers being placed in the assembly to mimic the rolling joint of the knee. This is the modification that needs to be made to the preexisting leg assembly. The inclusion of Figure 4 is not intended for determining which muscle contributes the most to the motion of the leg assembly, but more so for use as a visual reference of the mechanical formation of the knee joint.

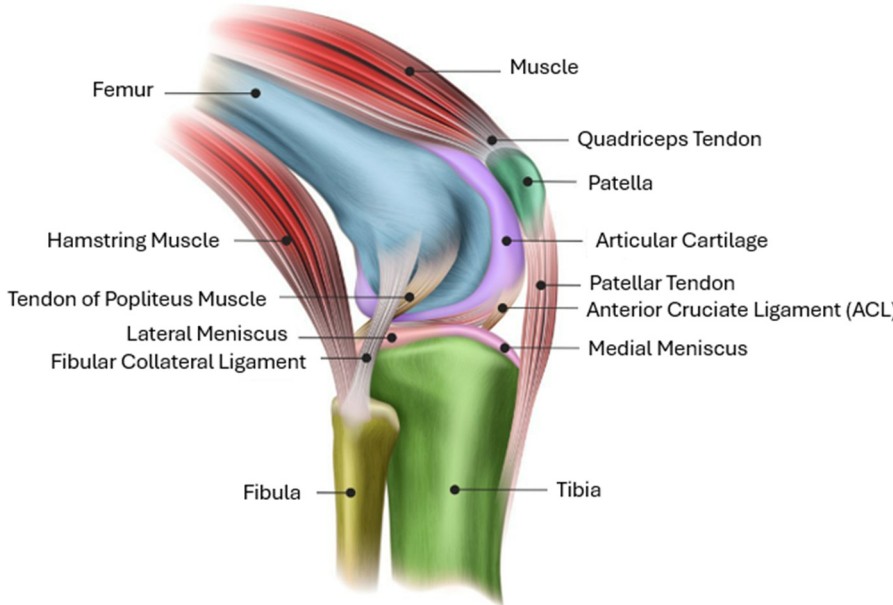

**Figure 4.** Anatomy of human knee joint [20].

Figures 5 and 6 present the reinforcement-learning (RL) Simscape model and the Simscape block diagram for the redesigned leg assembly, respectively. Figure 5 is the Simscape model that was used to conduct reinforcement-learning training using the redesigned leg assembly. Figure 6 is the block diagram in the Simscape model which assembles the

redesigned leg and allows for the joints to actuate based on the joint angles prescribed by the reinforcement-learning agent. Figure 7 depicts the variables which are controlled by the RL networks for the redesigned and original leg assemblies. In Figure 7, the variable q represents the joint angle for a specific joint, q1 corresponds to the hip joint, q2 to the joint above the knee segment, and q3 to the joint below the knee segment. The additional m subscript denotes joint angle calculated from the motion-capture data. The Simscape model consists of the redesigned leg model with three reference points at the hip, the midpoint of the second segment, and the ankle. Adjacent to the leg model are the mocap datapoints. The Simscape model features six different signals as seen in Figure 5: kinematic measurements of the leg model (meas) and mocap data (mocap meas), the action requested by the RL block (action), an observation of the state and performance of the redesigned leg assembly (observation), the reward generated by the performance of the leg assembly (reward), and a signal to end the current episode if the threshold distance between the leg model and mocap joints is met (isdone). The reinforcement learner interprets the observations and the reward that it received and performs a new action based on this feedback to maximize the reward it receives. The reward function governing the gait training is illustrated in Equation (1); the variables controlled by the RL agent for the redesigned and original RoboREHAB leg assemblies are depicted in Figure 7.

$$\text{Reward} = (x_k - x_{k,m}) - (y_k - y_{k,m}) - (x_a - x_{a,m}) - (y_a - y_{a,m}) \tag{1}$$

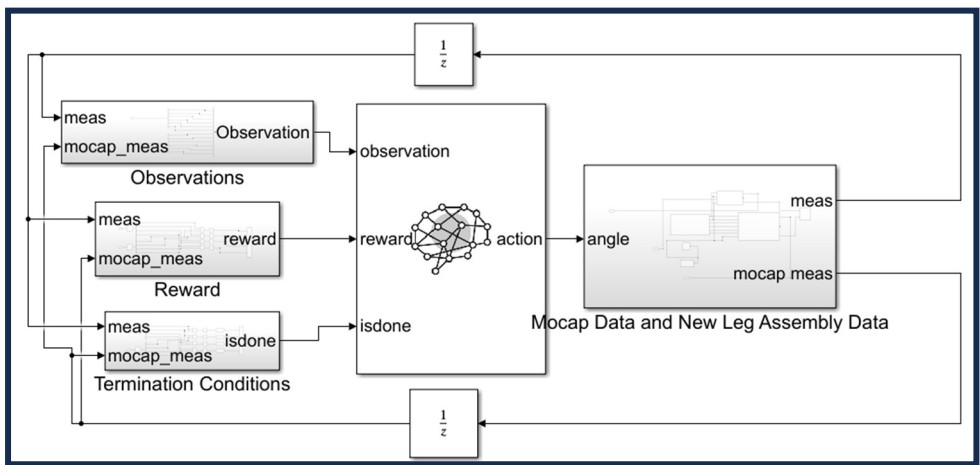

**Figure 5.** MATLAB/Simscape model of leg assembly.

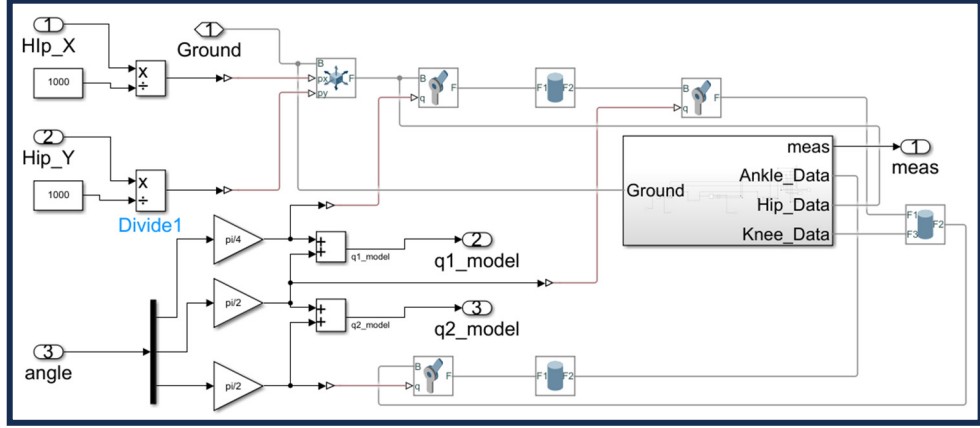

**Figure 6.** Leg subsystem for RoboREHAB leg assembly modified with rollers at knee.

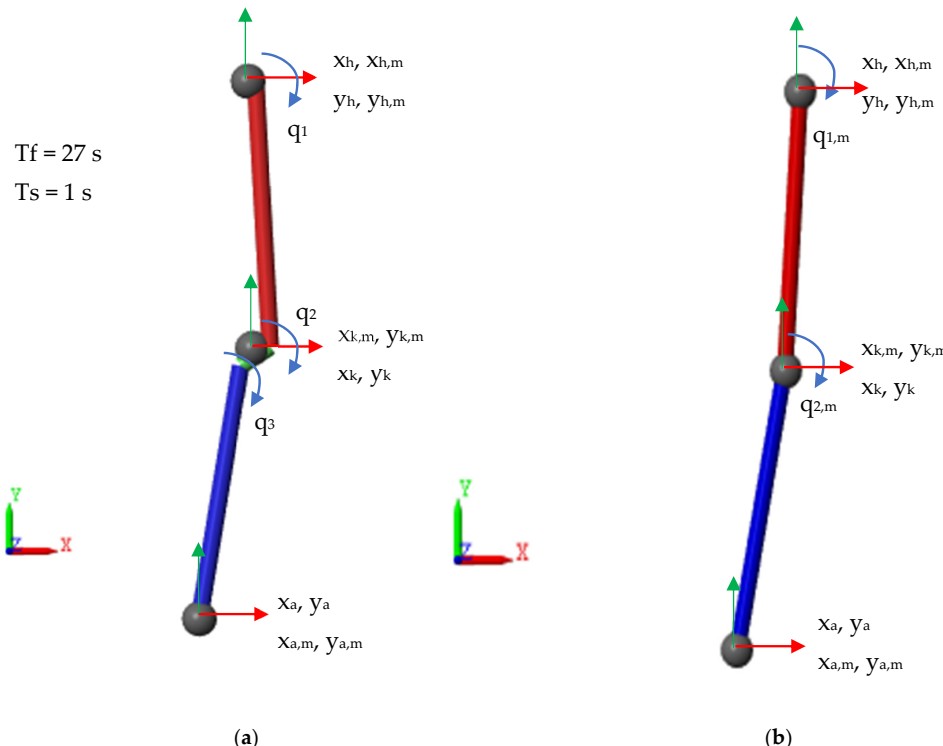

(**a**)                   (**b**)

**Figure 7.** Variables controlled in algorithm for (**a**) modified and (**b**) original RoboREHAB leg assemblies. Where $x_h$ and $y_h$ are the x and y positions of hip joint on leg model, $x_k$ and $y_k$ are the x and y positions of knee joint on leg model and $x_a$, $y_a$ are the x and y positions of ankle joint on leg model. $x_{h,m}$, $y_{h,m}$ are the x and y positions of hip joint from mocap data, $x_{k,m}$, $y_{k,m}$ are the x and y positions of knee joint from mocap data and $x_{a,m}$, $y_{a,m}$ are the x and y positions of ankle joint from mocap data. $q_1$ is the angle of rotation for first segment of the leg model, $q_2$ is the angle of rotation for second segment of the leg model and $q_3$ is the angle of rotation for third segment of the leg model. $q_{1,m}$ is the angle of rotation for the first segment of motion-captured limb and $q_{2,m}$ is the angle of rotation for second segment of motion-captured limb. $T_s$ is the sample time of the environment and $T_f$ is the final simulation time of the environment.

The logic flow for the algorithm is compiled below:

1. Initialize angles of the leg segments and load any previous experience if any exists.
2. For $t = 0$: $T_s$: $T_f$ sec.
3. Initialize state from observations signal as the first state in this sequence, retrieving joint positions and angles from the MATLAB leg model.
4. Obtain the action signal from the actor's current network.
5. Perform the action, obtain the new observation signal and reward signal, and determine if the episode ended prematurely by checking the 'isdone' block (does the distance between the leg model and mocap joints exceed the threshold?).
6. Store {observation, action, reward, newObservation, isdone} for empirical playback.
7. newObservation = observation.
8. action = prevAction.
9. Determine the q-value of the action from the empirical playback.
10. Update actor's and critic's networks.
11. If the new observation has satisfied the 'isdone' criteria, the episode is over. If not, continue with the next step.

The reinforcement-learning (RL) agent is trained for the redesigned and original RoboREHAB leg assemblies. The agent is trained for 10,000 episodes and the goal for the agent is to control the leg assembly by adjusting the joint angles of the segments to achieve a reward total of 0. During this training session, the code saves all agents that achieve a

reward of −30 or higher. If the agent achieves a reward total of −10, the training session stops early. The Simulink model has graph blocks to graph the joint trajectories of the hip, knee, and ankle joints of the RoboREHAB assembly, as well as the angular position histories. Joint-trajectory data were saved to a CSV file to be imported into an excel file to easily compare the results to the original RoboREHAB assembly and the mocap data. To ensure that the RL agent is properly trained, the angular position histories of the redesigned leg assembly and mocap data will be assessed to see how close the redesigned assembly is to the mocap data. Then, the joint-path trajectories will be compared for the redesigned and original leg assemblies and the mocap data to visualize the improvements of the redesigned assembly over the original design.

### 3. Results

The redesigned leg assembly and the original leg assembly are compared in Figure 8. The knee joint is incorporated into the link between the upper- and lower-leg sections and a close-up of the rollers for the knee joint is displayed to the right of the assembly. The result of the motion analysis conducted in SolidWorks is compared to the mocap data in Figures 9 and 10. SolidWorks motion analysis addressed feasibility of the redesigned leg assembly model; the SolidWorks model showed that it can replicate the movements and motions as gathered in the mocap data.

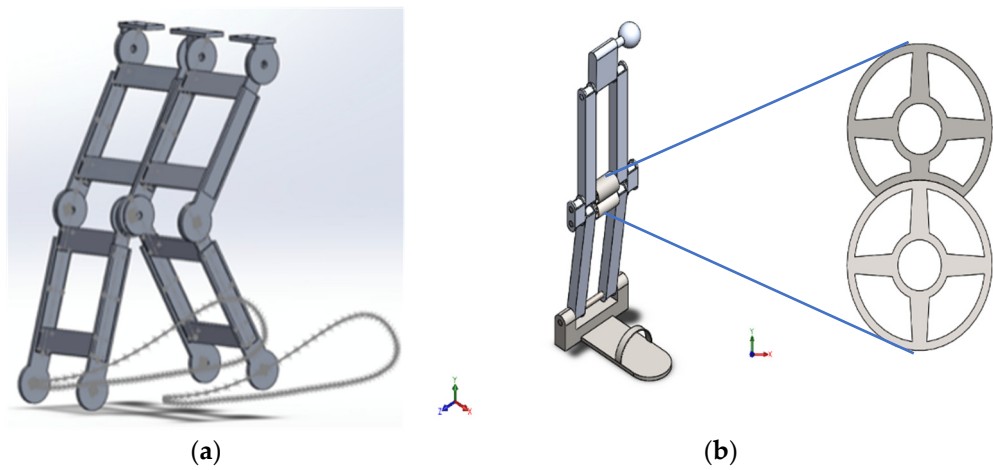

(**a**)  (**b**)

**Figure 8.** Comparison of configurations: (**a**) original RoboREHAB leg assembly and (**b**) redesigned leg assembly with extra DOFs and side view of roller mechanism.

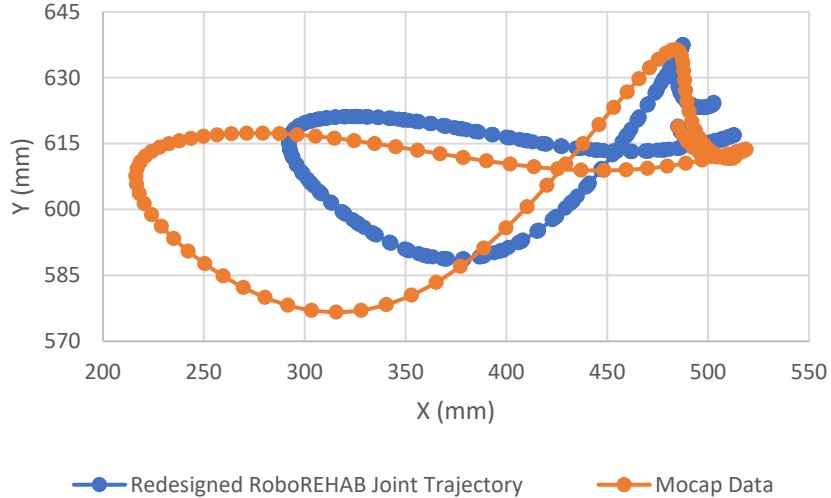

**Figure 9.** Comparison of knee-joint trajectories for mocap and redesigned model in SolidWorks.

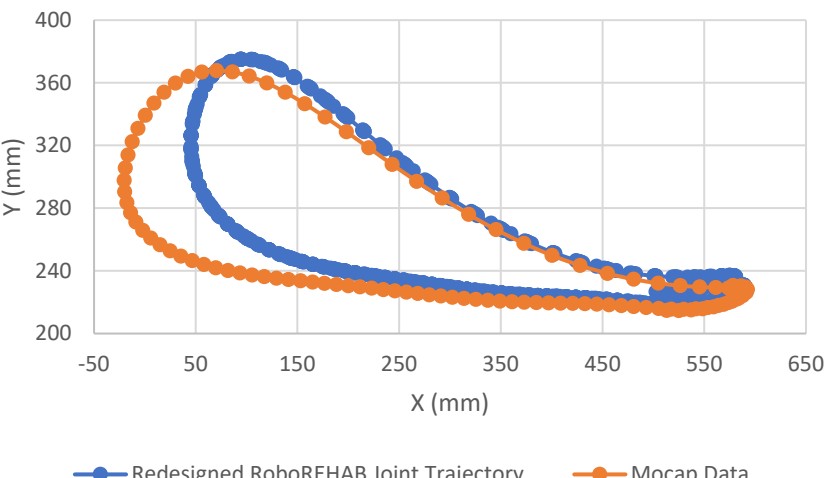

**Figure 10.** Comparison of ankle-joint trajectories for mocap and redesigned model in SolidWorks.

Figure 9, below, shows the knee movement of the newly redesigned and three-segmented leg assembly, in blue, overlayed onto the ideal knee-motion trajectory motion-capture data, in orange. This overlay of both plots validates the mechanical capabilities of the new leg assembly design, reiterating that the new knee joint can recreate the movement and ranges needed for ideal knee movement pattern during the gait cycle. To further ensure the functionality of the SolidWorks model, the ankle movement was also evaluated for motion and range. Figure 10 below shows the overlay of the SolidWorks model for the ankle joint movement, in blue, overlayed to the ideal mocap ankle-joint trajectory, in orange. From this data comparison, it is clear that the new redesign is fully capable of the motions and ranges needed to recreate an ideal gait cycle on the ankle joint.

While the SolidWorks model of the new redesign was used to validate the hardware aspect of the project, the MATLAB 2022a Simscape model was used to validate the software, programing, and machine-learning aspects of this project. In the Simscape model, the three-segmented leg model was trained in the reinforcement-learning block by the reward function illustrated in Equation (1). The trained leg model was then compared to the mocap data collected to validate the redesigned leg model in Simscape. The angular position histories for the mocap data and leg model are displayed in Figures 11 and 12, respectively. The Simscape simulation recreated the mocap data with less than a 5% deviation from the ideal mocap data. In Figure 11, the angular position of the leg segment connecting the hip and knee joints is shown by the red line. These data are directly compared to the angular position of the respective hip–knee segment of the mocap data. Much like the hip–knee segment, the knee–ankle segment simulation, shown in Figure 12, also shows the slight overshot of motion. This overshoot, like in the previous segment is still less than 5%, yet looks exaggerated when visualized in such a graphic. The initiation of these movements, marked at time 0, shows a larger-than-5% deviation. However, this shows that the machine-learning simulation can also somewhat naturally initiate a gait movement, also known as gait initiation.

The joint motions that the robots (leg assemblies in the model) learned are annotated in Figures 13–15. Figure 13 shows the hip-joint path movement throughout the gait cycle, with each axis representing relative movement to a reference frame. Three types of data point of the hip joint are plotted simultaneously to provide an intuitive visual of the results. In orange, the original movement of the hip joint, labeled as 'Mocap Data', extracted though motion-capture analysis, is plotted. This original motion-capture trajectory is considered the ideal movement of the hip joint through one repetition of the gait cycle. Next, the overlaying gray plot, labeled as 'Two Segment Simscape Model', shows the machine-learning algorithm replicating the hip motion with a Simscape model made of only two segments. Furthermore, the third plot, labeled as 'Three Segment Simscape Model', represents the reinforcement-learning algorithm replicating the hip movement of

a Simscape model made of three segments. All three of the data plots have converged to an identical hip trajectory, meaning that the reinforcement-learning algorithm was able to fully understand and replicate the hip motion of an ideal gait cycle. For this instance, the difference between two or three segments appears to be negligible.

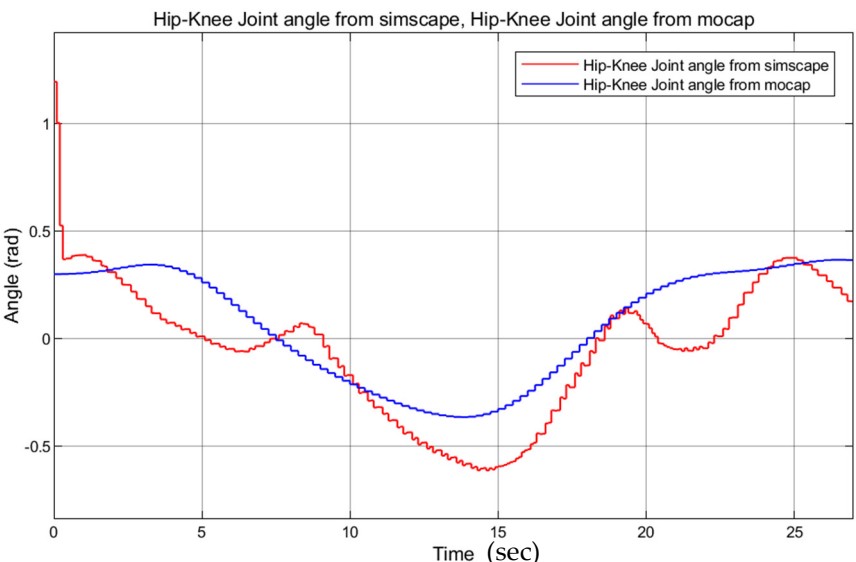

**Figure 11.** Angular position histories of hip–knee segment for Simscape model and mocap data.

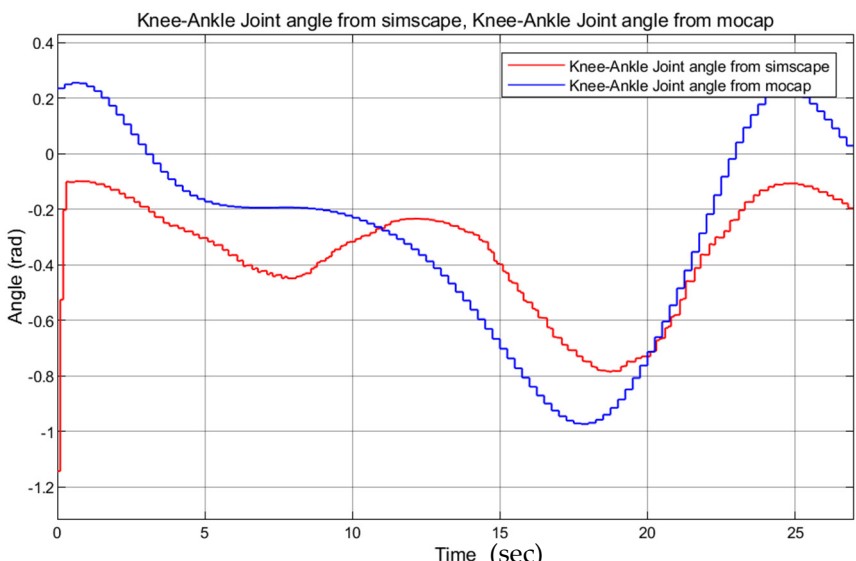

**Figure 12.** Angular position histories of knee–ankle segment for Simscape model and mocap data.

Figure 14 shows the overlay of the knee-joint movement paths of the motion capture, the Simscape model using two segments, and the Simscape model using three segments, respectively, following the same color convention as Figure 13 above. In this Figure, the two-segmented reinforcement-learning simulations, shown in gray, produced a deviation from the ideal knee-trajectory data, shown in orange. Next, focusing on the blue plot, the same reinforcement-learning algorithm replicated the knee-path trajectory, this time using the three-segmented leg assembly. Comparing these outputs shows that there is a significant difference between using a simulation with two segments compared to a simulation with three segments in replicating a knee trajectory implementing a machine-learning algorithm. The comparison of these two data plots, two-segmented and three-segmented, shows that the reinforcement-learning algorithm is better able to replicate an ideal and smooth knee-path trajectory using a three-segmented Simscape model. The maximum deviation from

the ideal path of the two-segmented simulation is 24.178 mm and the maximum deviation from the ideal path for the three-segmented simulations is 19.199 mm. In general, the simulation can replicate the horizontal elements of the movement better than the vertical elements of the knee-path trajectory.

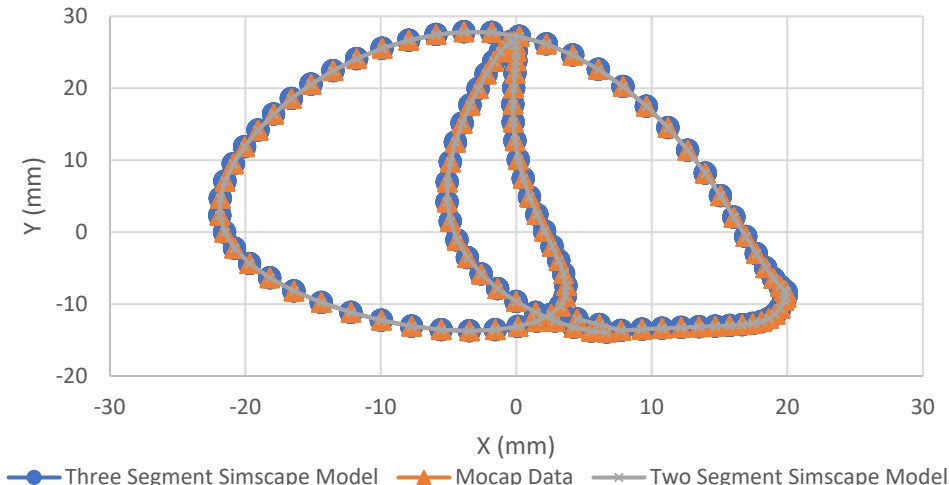

**Figure 13.** Hip-path trajectory from the mocap data and Simscape models of two/three segments.

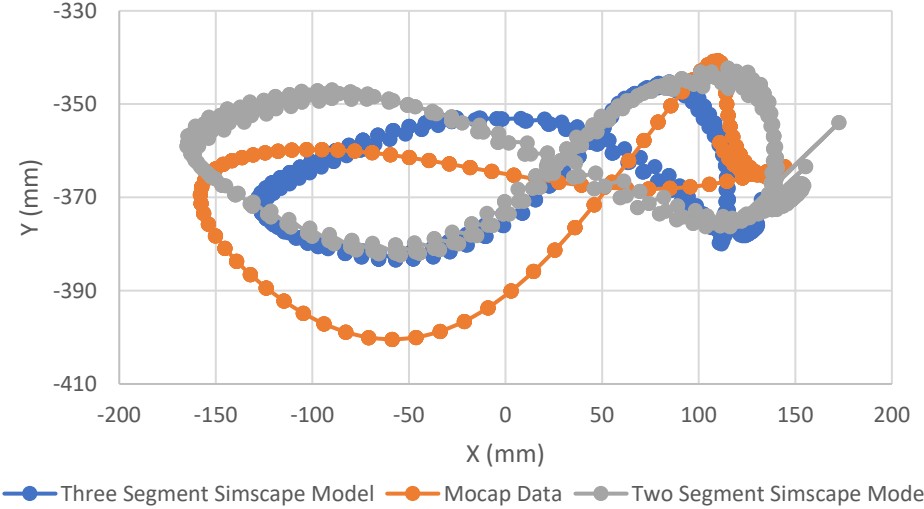

**Figure 14.** Knee-path trajectory from the mocap data and Simscape models of two/three segments.

Figure 15, once again, shows an overlay of joint trajectory, specifically, the three ankle trajectories with their respective colors. While errors from the hip and knee joints tend to increase as they propagate downwards though the kinetic chain, the results of the simulation are pleasing. The reinforcement-learning algorithm was able to replicate an ankle-motion trajectory using the two-segmented and the three-segmented Simscape models. Much like the knee trajectory, the two-segmented ankle trajectory is less ideal than the three-segmented results. This further validates the idea that the addition of the knee segment does in fact aid the smoother and more accurate replications of gait motion. While the errors of the knee trajectory propagate a larger error of the ankle, the reinforcement-learning algorithm was able to compensate significantly. The error of the two-segmented simulation plot was 72.084 mm compared to the error of the three-segmented simulation, 51.177 mm.

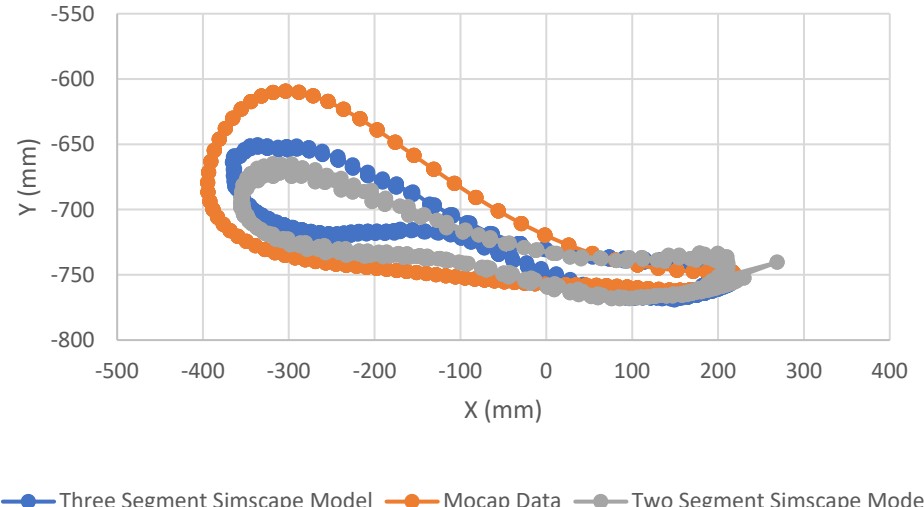

**Figure 15.** Ankle-path trajectory from the mocap data and Simscape models of two/three segments.

## 4. Discussion

The angle-position histories of the Simscape model and the mocap data shown in Figures 11 and 12 share a similar profile but the Simscape model has variations in the angle at various points of the simulation, which cause deviations in the path trajectories of the knee and ankle joints as shown in Figures 14 and 15. The path trajectories of the redesigned three-segmented leg assembly are close to the mocap data but feature deviations due to the angle difference between the mocap data and the leg assembly. The sources of the angle difference could be due to the lengths of the leg assembly segments remaining constant and the fact that the motion in the z-direction is not considered in our current work. However, the maximum deviations for the ankle and knee joints (8.3%, 4.8%) are smaller for the three-segment assembly than the two-segment assembly (11.8%, 6.1%) but maintained a similar profile to the mocap trajectory data. This shows promise for the redesigned leg assembly and can be further improved by resolving the deviation. A reason for this deviation could be due to the segment lengths remaining constant throughout the simulation. The mocap data reveal small shifts in the distance between the joints every single frame, which demonstrates small changes in the position of the joints that cannot be captured by having the leg segments at a constant length. These changes in segment length of the mocap data are caused by movement into the unmeasured axis, the z axis. If segment-length compensations were permitted by the Simscape model, the joint-path trajectories of the redesigned leg assembly could be nearly exact to the mocap data. Further work would therefore involve changing the leg segments to feature variable lengths using adjustable spring stiffness or other equivalent mechanisms and comparing the joint trajectories obtained to the mocap data. Implementation of variable-length leg segments in the future might be the key to replicating a more natural human gait motion and the exact replication of the joint trajectories.

This work's impact extends to predicting and correcting gait motion and to joint angle and torque derivation. Such derived data could be used as a tool to assess the degree of paralysis that patients are suffering from. Furthermore, this derived data can also serve to prescribe specialized treatments to address specific problem areas.

One of the promising applications of such data would be critical in mapping the path trajectory to the joints to develop smart implants with triboelectric patterns. Joint implants, such as for the knee, hip, and ankle, would therefore be able to signal their remaining life. Such a smart implant would alert physicians for replacement before it fails and injures the patient. The path trajectory mapping can also be applied to characterize asymmetry during gait by comparing relative motion in the left–right angle diagram [21] on the hip, knee, and ankle joints, respectively. This method may thus facilitate physicians' decisions in the

early diagnosis and in the evaluation of disease progression in diseases such as Parkinson's disease [22].

## 5. Conclusions

In this work, a redesign of the leg assembly for RoboREHAB was presented that aims to improve the joint movements of the knee and hip joints that can be achieved by traditional leg assemblies. SolidWorks motion analysis and Simscape analysis demonstrated the feasibility of the redesigned leg assembly. Reinforcement learning was applied to a Simscape model that utilized the redesigned leg assembly. The policy that the network learned while using the new leg design produced angular-position histories and path trajectories that closely reflect the results from the mocap data when compared to the original two-segmented RoboREHAB leg assembly. Results of the validation process showed that, based on the shape and size of the reproduced trajectories, a smooth and more natural gait was learned by the reinforcement-learning algorithm using the three-segmented Simscape model. The learned trajectories, while still improvable, will serve as a foundational reference to diagnose and evaluate normal gait deviations for patients. Therefore, the key result is that a redesigned RoboREHAB, with a three-segmented linkage provides a framework that is better for rehabilitating and recreating human joint movements used in robotic rehabilitation applications. This key finding opens the door to monitor patient progress objectively by direct comparison of the patient's current gait and the natural human gait, allowing for specialized treatment of the muscles or neural connections responsible for the abnormal gait. The learned trajectories could be improved by using variable-length segments for the leg assembly or incorporating the z-direction motion from the motion-capture data to train the reinforcement-learning agent.

**Author Contributions:** Conceptualization, J.A., C.-H.G. and Y.T.W.; data curation, J.A.; formal analysis, J.A.; investigation, J.A. and A.Y.; methodology, J.A., C.-H.G. and A.Y.; project administration, C.-H.G. and A.Y.; resources, C.-H.G. and Y.T.W.; software, J.A.; supervision, C.-H.G.; validation, C.-H.G. and A.Y.; writing—original draft preparation, J.A.; writing—review and editing, C.-H.G., A.Y. and Y.T.W. All authors have read and agreed to the published version of the manuscript.

**Funding:** This project has been partially supported by the Capstone Senior Design teams in the Department of Mechanical Engineering at the University of Texas at Tyler since 2019. The authors thank their contributions to building a prototype of the RoboREHAB (robotic walking training device).

**Institutional Review Board Statement:** Not applicable.

**Informed Consent Statement:** Not applicable.

**Data Availability Statement:** The data presented in this study are available on request from the corresponding author. The data are not publicly available due to privacy.

**Conflicts of Interest:** The authors declare no conflicts of interest.

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
