# Peer review of "Redesign of Leg Assembly and Implementation of Reinforcement Learning for a Multi-Purpose Rehabilitation Robotic Device (RoboREHAB)"

_applsci, doi:10.3390/app14020516_

Round 1

Reviewer 1 Report

Comments and Suggestions for Authors

Authors have attempted an interesting problem by redesigning the current design. These are my observations. (a) The analytical content of the paper is poor. (b) How the data was captured and the sensors utilized? The simulation framework used should be explained well. 

Abstract must reflect the improvement achieved using some numerical result. Claims without numerical values have less impact.

How reinforcement learning model is implemented? Please explain the content. The discussion on actuators in the developed model is not clear. 

It seems that the proposed design does not have any power input for gait improvement. In that case, how the model is useful is not clear. 

Please elaborate Fig. 7 to 9.

The symbols q1,m  and q2,m are not clear in Fig 9. Please explain 

Comments on the Quality of English Language

ok

Author Response

Reviewer 1 (Revisions are highlighted in yellow in the revised document)

  1. Authors have attempted an interesting problem by redesigning the current design. These are my observations. (a) The analytical content of the paper is poor. (b) How the data was captured and the sensors utilized? The simulation framework used should be explained well.
    • Thank you for your feedback. We added revisions to the document that improved the paper's analytical content by explaining the improvements of the three-segment assembly over the two-segment assembly. We also added contents to the methodology explaining how the data from the Simulink model was collected and compared to the original assembly design and the mocap data (see section 2. Design and Methods for details).
  2. Abstract must reflect the improvement achieved using some numerical result. Claims without numerical values have less impact.
    • We appreciate your feedback on the abstract, we have added numerical results to the abstract and have highlighted the revision in yellow.
  3. How reinforcement learning model is implemented? Please explain the content. The discussion on actuators in the developed model is not clear.
    • Thank you for your feedback on this detail. Revisions have been made to the manuscript that further explain how the reinforcement learning is used to train the agent to adjust the joint angles of the redesigned RoboREHAB leg assembly to follow the motion capture data (see section 2. Design and Methods for details).
  4. It seems that the proposed design does not have any power input for gait improvement. In that case, how the model is useful is not clear.
    • Thank you for your input. The output of the reinforcement learning agent in the Simscape model provides the signal which adjusts the joint angles of the leg segments, this signal can be equated to controlling the torque of the motors at each joint. This explanation has been added to the revisions which are highlighted in yellow.
  5. Please elaborate Figures 7 to 9 (Figures 5 to 7 in the revised version).
    • We combined Figures 1 to 3 shown in the original manuscript with Figure 1 in the revised version according to the reviewer’s suggestion. As a result of this, figures were renumbered in the revision. Figure 5 is the Simulink model used to conduct reinforcement learning training using the redesigned leg assembly. Figure 6 is the block diagram in the Simulink model which assembles the redesigned leg and allows for the joints to actuate based on the joint angles prescribed by the reinforcement learning agent. Figure 7 shows the variables controlled by the RL network for the redesigned and original leg assembly. Revisions have been made to the design and methods section to provide more explanation for the figures (revisions are highlighted in yellow in the revised version).
  6. The symbols q1,m and q2,m are not clear in Fig 9 (Figure 7 in the revised version). Please explain
    • Thank you for your comment, the variable q represents the joint angle for a specific joint, q1 corresponds to the hip joint, q2 for the joint above the knee segment and q3 for the joint below the knee segment. The additional m subscript denotes joint angle calculated from the motion capture data. A revision has been made to Figure 9 (Figure 7 in the revised version) to resolve any confusion.

Reviewer 2 Report

Comments and Suggestions for Authors
  1. Figures 4, 6, and 10(a) cited in this paper have poor image quality and appear blurry.

  2. Figure 6 indicates that the human knee joint is very complex, and the original structure cannot accurately mimic the trajectory of the human lower limb. Therefore, the paper suggests adding links to increase degrees of freedom and better mimic the trajectory of the human lower limb. However, the paper does not explore how the human knee joint substantially contributes to the improvement, such as analyzing which muscle parts of the human knee joint play a dominant role in knee movement. Additionally, it does not discuss how utilizing the trajectory of the human knee joint motion can assist in the structural design of the exoskeleton knee joint.

  3. Is there a basis for the optimization by adding a link? Are there other better optimization solutions?

  4. Figures 16 and 17 compare the ankle trajectory paths between motion capture data and simulated scene model. The trajectories seen in the images show little difference between the original and improved paths. As for the knee joint path simulation, the paper states that the maximum deviation from the ideal path for the original and improved structures is 24.178 mm and 19.199 mm, respectively. The errors for the ankle joint in the original and improved structures are 72.084 mm and 51.177 mm, respectively. The paper concludes that the improved structure has a better effect. However, is the maximum deviation of the path a widely accepted evaluation standard in this field? Are there other evaluation indicators to represent better results after optimization?

Comments on the Quality of English Language

No

Author Response

Reviewer 2 (Revisions are highlighted in cyan in the revised document)

  • Figures 4, 6, and 10(a) cited in this paper have poor image quality and appear blurry.
    • Thank you for the feedback, the pictures have been revised. In the revised version, you can see the improved quality of Figures 2, 4, and 8 (a); figures were renumbered in the revision.
  • Figure 6 indicates that the human knee joint is very complex, and the original structure cannot accurately mimic the trajectory of the human lower limb. Therefore, the paper suggests adding links to increase degrees of freedom and better mimic the trajectory of the human lower limb. However, the paper does not explore how the human knee joint substantially contributes to the improvement, such as analyzing which muscle parts of the human knee joint play a dominant role in knee movement. Additionally, it does not discuss how utilizing the trajectory of the human knee joint motion can assist in the structural design of the exoskeleton knee joint.
    • We appreciate your feedback on the methodology of this manuscript. The inclusion of Figure 6 (Figure 4 in the revised) is not to determine which muscle contributes the most to the motion of the leg assembly but more so to use as a visual reference of the mechanical formation of the knee joint. It is included to help explain why additional degrees of freedom need to be added at the knee joint of the assembly. The use of joint path trajectories aids in the development of the reinforcement learning agent. This is used to actuate the joints of the leg assembly. Revisions (highlighted in cyan) have been made to the manuscript to reinforce this point.
  • Is there a basis for the optimization by adding a link? Are there other better optimization solutions?
    • This is an exceptionally good question, for this work we are specifically concerned about how adding an additional link can improve the gait trajectory of the RoboREHAB leg assembly. We explained why we added an additional link on the knee joint in the manuscript as follows: “The original design of the RoboREHAB leg assembly shown in Figure 2 does not accurately replicate the path trajectory of the knee and hip joints shown in Figures 1b and 1c due to a lack of degree of freedom (DOF) in the assembly.” “This results in increasing DOFs such as pitch moment on the knee joint during the gait motion simulations so that it produces better path trajectories on the joints than the original design at which only swing motion was allowed on the knee joint.” We are not aware of better optimization solutions but most of the lower extremity leg assemblies only have one degree of freedom at the knee joint.
  • Figures 16 and 17 compare the ankle trajectory paths between motion capture data and simulated scene model. The trajectories seen in the images show little difference between the original and improved paths. As for the knee joint path simulation, the paper states that the maximum deviation from the ideal path for the original and improved structures is 24.178 mm and 19.199 mm, respectively. The errors for the ankle joint in the original and improved structures are 72.084 mm and 51.177 mm, respectively. The paper concludes that the improved structure has a better effect. However, is the maximum deviation of the path a widely accepted evaluation standard in this field? Are there other evaluation indicators to represent better results after optimization?
    • Thank you for your feedback. The use of maximum deviation is used in this instance to highlight the difference between the original RoboREHAB design and the new design with the additional links and rollers. The original design of the RoboREHAB leg assembly shown in Figure 2 does not accurately replicate the path trajectory of the knee and hip joints shown in Figures 1b and 1c due to a lack of degree of freedom (DOF) in the assembly. The redesigned leg assembly results in increasing DOFs such as pitch moment on the knee joint during the gait motion simulations so that it produces better path trajectories on the joints than the original design at which only swing motion was allowed on the knee joint.

Reviewer 3 Report

Comments and Suggestions for Authors

1. It is recommended to consolidate multiple images grouped together and assign sequential numbering. Subsequently, provide brief captions below each consolidated image, as illustrated in the example below:

 2. It is advised to reconsider the layout of images to minimize overlap between text and graphic borders, which can lead to blurriness and non-uniform presentation, as indicated by the red box in the example below:

3. In the discussion section, a more in-depth explanation of the sources of angle differences in the model, such as specific joints or motion directions involved, is recommended to enhance readers' understanding of the issue.

4. In the conclusion section, it is suggested to appropriately emphasize potential improvements in learning trajectories through reinforcement learning. This could involve adjusting model parameters, optimizing algorithms, or increasing training data.

5. In the conclusion section, it is advisable to highlight the advantages of the redesigned RoboREHAB in monitoring patient rehabilitation progress. Emphasize that the redesigned system opens up possibilities for more objective monitoring of patients during the rehabilitation process, enabling a comprehensive evaluation of their progress.

Comments on the Quality of English Language

Minor editing of English language required.

Author Response

Reviewer 3 (Revisions are highlighted in green in the revised document)

  1. It is recommended to consolidate multiple images grouped together and assign sequential numbering. Subsequently, provide brief captions below each consolidated image, as illustrated in the example below:
    • Thank you for your feedback. We have consolidated figure 1 in the introduction and highlighted in the revisions.

  1. It is advised to reconsider the layout of images to minimize overlap between text and graphic borders, which can lead to blurriness and non-uniform presentation, as indicated by the red box in the example below:
    • We appreciate your advice on the layout of the figures, the text and figures have been modified to provide a more uniform presentation, and the blurriness of some of the images has been resolved in the revision (highlighted in green).

  1. In the discussion section, a more in-depth explanation of the sources of angle differences in the model, such as specific joints or motion directions involved, is recommended to enhance readers' understanding of the issue.
    • Thank you for your feedback, we have stated in the discussion that the sources of the angle difference could be due to lengths of the leg assembly segments remaining constant and the fact that the motion in the z-direction is not considered in our current work.

  1. In the conclusion section, it is suggested to appropriately emphasize potential improvements in learning trajectories through reinforcement learning. This could involve adjusting model parameters, optimizing algorithms, or increasing training data.
    • We appreciate your comment, a revision has been added to the conclusion highlight how the use of variable length segments or the z-axis motion capture data can improve the learned trajectories (revisions are highlighted in green).

  1. In the conclusion section, it is advisable to highlight the advantages of the redesigned RoboREHAB in monitoring patient rehabilitation progress. Emphasize that the redesigned system opens up possibilities for more objective monitoring of patients during the rehabilitation process, enabling a comprehensive evaluation of their progress.
    • Thank you for your feedback, a revision has been made to the conclusion mentioning how the RoboREHAB can not only provide rehabilitation training but also be used as a diagnostic tool. By analyzing the patient’s gait with RoboREHAB, specialized treatment can be prescribed to exercise the muscles or neural connections that are causing the abnormal gait (revisions are highlighted in green).

Round 2

Reviewer 3 Report

Comments and Suggestions for Authors

The author responded well to my suggestion.